# Crafting Metal Surface Morphology to Prevent Formation of the Carbon–Steel Interfacial Composite

Yuanhuan Zheng [1], Siok Wei Tay [1] and Liang Hong [2,*]

1   Institute of Materials Research and Engineering, 2 Fusionopolis Way, Innovis, #08-03,
    Singapore 138634, Singapore
2   Department of Chemical & Biomolecular Engineering, National University of Singapore,
    4 Engineering Drive 4, Singapore 117585, Singapore
*   Correspondence: chehongl@nus.edu.sg

**Abstract:** We created a coke-repellent inner surface in a stainless steel (SS-321) tube using an enhanced chemical etching tactic. A water-borne etching solution was formulated by combining an ion sequestering ligand (L), hydrogen peroxide (H), hydrochloric acid (C), and a stabilizing agent (E or N). Three etchants, LHC, LHC-E, and LHC-N, were therefore formulated, respectively. The coke-repellent metal surfaces achieved by these etchants all show a characteristic topographic pattern on a micron scale, specifically with grooved spherulite and ridge-like topographic patterns. Fundamentally, these two topographic patterns prompt overhead micro turbulence fields whose agitation mitigates the surface entrapment of aromatic hydrocarbon flocs generated from the overhead lubricant. The surface entrapment of flocs is the crucial step to trigger coke growth. The coke repellency was assessed by placing an SS-321 tube filled with a lubricant in a heat soak. It was found that the topographic pattern and its surface roughness level have opposite effects on coke development. Hence, the three etchants give rise to different coke-resilient surfaces. Moreover, the plug flow rate of the etchant also affects the anti-coking performance, exhibiting an optimal flow rate that offers the highest coke-proof efficacy.

**Keywords:** coke resistance; metal surface topographic pattern; stainless steel tube; chemical etching; formulation of etchant; lubricant

## 1. Introduction

Coking is a severe challenge to the industries that utilize and manufacture hydrocarbons at high temperatures [1,2]. Coking originates from the thermal decomposition of hydrocarbons, particularly the no-slip surface liquid film that produces aromatic species to initiate coking [3]. The growing of aromatics commonly results in deposition on hot metal surfaces of different devices, such as reactors, pipes, valves, and nozzles, as well as the metallic sites of the supported catalyst. The main adverse effects of coking are coil clogging [4], catalyst deactivation [5], hindering heat transfer [6], and causing corrosion and erosion [7]. Hence, many efforts have been undertaken to bring coke formation and deposition under control in the oil refinery, energy, and transportation industries by various measures. Such measures typically include adding inhibitors and antioxidants to fuels [8,9] or laying an anti-coking coating or a passive layer on the metal surfaces of interest [10,11]. These measures reduce the kinetics of thermal cracking, leading to polyaromatics in the oil bulk and hence prevent the proliferation of hydrocarbon flocs. Structurally, the flocs are aggregates or fusions of aromatic compounds. The fastening of these flocs, mainly coming from the surface oil film and the bulk as well, to the metal surface of the device facilitates their conversion to carbonaceous particles [12,13] because the metal part is hotter and possesses dehydrogenation catalytic activity owing to group VIII elements [14]. In addition, surface roughness promotes coking propensity [15]. Besides temperature, coking is also profoundly affected by the viscosity of hydrocarbon liquid since it is related to the sliding of the surface oil film and heat transfer coefficient [16].

This study focuses on the coking problem existing in lubricating oil tubes of automotive power systems, where chronic exposure to heat soak, being the residual heat radiated from an engine after it is stalled, leads to decomposition of lubricating oils in pipes and nozzles, and therefore, the deposition of carbonaceous solid [17]. In general, oxidation and thermal reactions cause the formation of coke particles. In a closed lubricating oil system, the second origin prevails because today's aviation engines operate at significantly high temperatures to enable engines to travel further distances. Therefore, the aviation industry is more interested in the high thermal stability of engine oils [18], requiring the addition of stabilizers. Similarly, the coking residence time (RT) is affected by the oil–metal interfaces. Four scenarios are considered: an oil film (short RT), uneven wetting (variable RT), an oil layer (long RT), and a change in oil flow behavior and metal surface states (variable RT) [19]. The coke particles suspended in lubricant could result in bearing wear, run-out, and stick due to insufficient lubrication [20] and an erosive effect when they enter the hydrocarbon stream, where these particles act like abrasives on the other metallic parts of the system. As highlighted above, the current approaches to inhibit coking have pros and cons (*i*) The addition of a coking inhibitor to oil is limited to applications with a finite heating duration and at a temperature below the decomposition point of oil because the significant type of inhibitor is a free radical scavenger, which will lose performance after it is converted to an inactive free radical species or badly condenses with other organic fragments. (*ii*) Laying a ceramic passive coating on the inner surface of a lubricant tubing is inconvenient because of the small internal diameter and the risk of damaging bearings or valves once ceramic particles drop out from the coating due to a mismatch of thermal expansion coefficients between substrate and coating in a repeated heating and cooling process. Meanwhile, developing an inert barrier layer through surface bonding could avoid the adhesion issue, for example, by forming an overlying metal sulfide film [21] and a weld-mounted overlaid layer [22]. However, with this design, it is difficult to achieve a uniform and complete coverage over the inner surface of the metal tubing since this approach is usually location-selective because the inner surface bears numerous defects caused by extrusion to make the tube.

This study tackled the coking problem of a narrow stainless steel tubing conveying aerospace lubricant, which is exposed to a heat soak (~280–300 °C) in a cycling manner, by an alternative approach to those invented and used thus far. Our approach utilizes the acid chemical etching principle to formulate a quaternary chemical etching solution (acid, oxidizing agent, ligand, and stabilizer) to carry out a controlled surface etching. Conventionally, the chemical etching technique is employed to create fine trenches or holes on metal sheets and plates, e.g., printed circuit heat exchanger channels [23]. Typically, an etching solution consists of an acid (e.g., HCl or/and $H_3PO_4$), an oxidizing agent (e.g., $H_2O_2$ or $FeCl_3$), a ligand (e.g., lactic acid, $F^-$ or oxalic acid), and a free radical stabilizer to inhibit fast decomposition of hydrogen peroxide [24,25]. These constituents synergize the fundamental aquatic reaction equilibria to satisfy several demands of etching, e.g., regulating etching rate and depth, averting hydrolysis, smoothening the etched surface, and stabilizing the etching bath. According to the etching chemistry, this study's etching solution permits polishing the rough surface of the inner tubing to remove micro defects and coarse oxide top layer, creating a particular microstructure characterized by a grooved spherulite or a ridge-like topographic pattern on a submicron scale. Both micro-structures demonstrate coke-inhibiting attributes through extended contact with an aerospace lubricant at 300 °C. Theoretically, each topographic pattern leads to a turbulent heat flux [26], and therefore, instantaneous thermal emissivity and temperature differences [27] cross each patterned loop. As a result, a thin liquid layer is stimulated, consisting of numerous mini turbulence fluid circles over the tube wall [28]. As a result, the dynamic fluid state at the oil–metal interface driven by continuous variation in mass densities in the micron range pushes away aromatic flocs from sticking to the metal surface. In the meantime, this anti-coking mechanism does not influence thermal reactions that form aromatic flocs in

the lubricant bulk. The mechanism works because the conversion of aromatic flocs to coke particles is far slower in the oil bulk than on the metal surface.

## 2. Materials and Methods

### 2.1. Materials

Hydrochloric acid (HCl) (37%, Sigma Aldrich, St. Louis, MO, USA), hydrogen peroxide ($H_2O_2$) (30–32 wt %, Merck, Darmstadt, Germany), nitric acid ($HNO_3$) (65%, Sigma-Aldrich), (s)-lactic acid (~90 wt %, Merck), etidronic acid ($C_2H_8O_7P_2$, 60 wt % aqueous solution, Sigma-Aldrich), 1,5-naphthalene disulfonic acid (powder, >96.5%, Sigma-Aldrich), acetone (AR grade, Kanto Chemical Co. Inc., Tokyo, Japan), isopropanol (ACS and HPLC grade, J.T. Baker, Phillipsburg, NJ, USA), and methanol (HPLC grade, J.T. Baker) were used as received. Deionized (DI) water (18 mΩ·cm) was obtained from the Milli-Q IQ 7003 pure lab water purification system. Commercial aerospace lubricant (Mobil Jet Oil II) and the stainless steel tubes (SS-321 grade, od = 6.41 mm, id = 4.73 mm, Plymouth Tube Company, Salisbury, MD, USA) were provided by Honeywell.

### 2.2. The Controlled Chemical Etching of the SS-321 Tube

The SS-321 tubes were cut into pieces with a length of 9 cm each and chamfered at both ends for use as a specimen for testing. The tubes were ultrasonicated in acetone for 10 min and dried in ambient conditions. The chemical etchant solution was formulated by mixing 9 mL $H_2O_2$, 27 mL HCl, 15 mL lactic acid ($C_3H_6O_3$), and 0.6 g etidronic acid solution or 0.088 g 1,5-naphthalene disulfonic acid with 3 mL DI water and stirring for 2 min; the content of etidronic acid is 0.57% and that of 1,5-naphthalene disulfonic acid is 0.14%. Other etchant solutions were prepared for comparison. An SS-321 sample tube was connected by two silicone tubes, one at each end. An etchant was then driven through the sample tube by a peristaltic pump operated at 150 RPM for 30 min, as illustrated in Figure 1. The etching process produces heat and foams. After completing the chemical etching procedure, the treated SS-321 tube was rinsed with a copious amount of tap water and soaked in DI water at 70 °C with sonification for 15 min. The tube was rinsed with DI water and dried in an oven at 80 °C for 20 min. Similarly, the other sample tubes were also prepared by varying etchant composition or etching conditions.

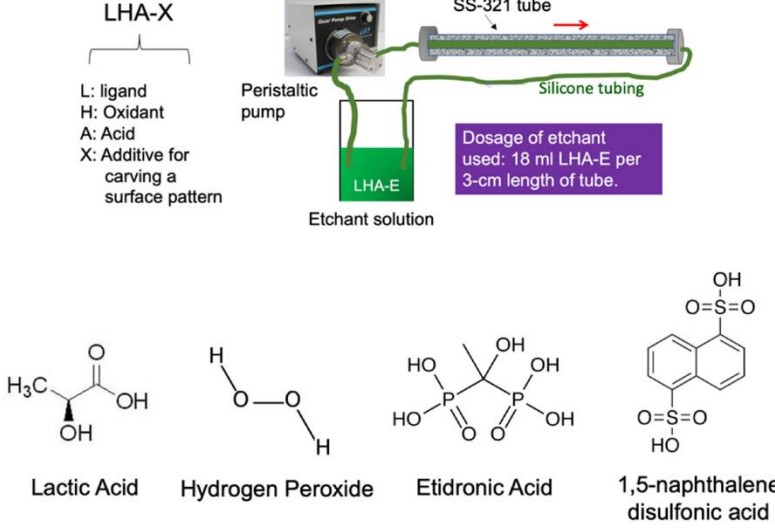

**Figure 1.** Schematic of the setup to conduct chemical etching using the plug flow reactor (PFR) mode (upper). The structures of the L (lactic acid), H (hydrogen peroxide), and X (etidronic acid or 1,5-naphthalene disulfonic acid) are listed.

### 2.3. The Assessment of Coking Resistance of the Inner Surface of the SS-321 Tube

Lubricant (Mobil Jet Oil II) was filled to the brim of a specimen tube and then sealed firmly in the tubing by steel nut/cap sets. The oil-loaded tube was then heated in a tube furnace using a ramp of 5 °C/min to 300 °C, dwelling for 12 h, and cooling down to room temperature at 5 °C/min. The heated lubricant was drained from the tube and refilled with fresh oil to conduct the second round of heating. This testing process was repeated for several cycles (3 to 5) to simulate the temperature alternation of heat soak. Another test was to prolong the dwelling to 56 h for each process. After the assessment, the tube discharged was immersed in isopropanol for 30 s with ultrasonication to wash away soluble species attached to the inner surface. The tubing was subsequently cross-sectioned with a junior hacksaw to expose the inner surface for microscopic examinations.

### 2.4. Characterization and Analysis

The internal surface roughness of an SS-321 sample tube after being treated by a specific acid etching formula was examined using a surface profilometer (Kla Tencor P10). The interior surface, before and after the coking assessment, was characterized on a scanning electron microscope (SEM) (JOEL 6360LA), and the surface composition of an etched surface was determined by energy dispersive spectroscopy (EDS) (Oxford Instruments X-Max). In addition, the composition of a tested lubricant oil sample and the particle sizes of aromatic flocs collected from the oil sample were analyzed by UV–Vis spectroscopy (Shimadzu UV-2501PC), dynamic light scattering analysis (Brookhaven Instrument Corporation NanoBrook 90plus Zeta Particle Analyzer), and liquid chromatography mass spectroscopy (Shimadzu LCMS-8030) equipped with a separation column (Shim-pack XR-ODS). The samples were prepared for the latter two analyses by adding a small amount of the oil sample (0.4 mL) in 2-propanol (4 mL). Raman spectroscopy examination on the coke residue collected from the metal surface of interest was conducted on the equipment (Renishaw inVia Raman Microscope). After chemical etching, the change in mechanical properties of the SS-321 tube was assessed on an Instron mechanical tester (Instron 5569 Universal Testing Machine).

## 3. Results

### 3.1. Thermal Decomposition of the Designated Aerospace Lubricating Oil

The thermal cracking extent of the aerospace lubricant sealed in an SS-321 tube was examined by placing it in a furnace to simulate heat soak. In this testing system, not just temperature but also the confined oil and the curved metal surface impact the outcome of the examination. Although the oil is a proprietary multi-component mixture, the literature [29] and the FT-IR spectrum of the fresh lubricant (Figure S1) suggest that its main constituents are m-phthalates, aliphatic esters, and phosphates with long alkyl chains. The spectrum serves to assist with the understanding of the thermal degradation and subsequent carbonization behaviors of the oil rather than to determine the precise components.

We first looked at the thermal decomposition profile of the oil (Figure 2). Three oil samples having different heating histories start to decompose at about 280 °C but show different thermal gravimetric (TG) mass loss trajectories during the removal of the last 30% of the mass. The fresh oil sample was first wiped out in air purging because the other two with different prior heating histories contained thermally stable aromatics. We further examined the change in composition near the decomposition temperature (Figure 3). The oil samples were heated at different temperatures under Ar in glass vials.

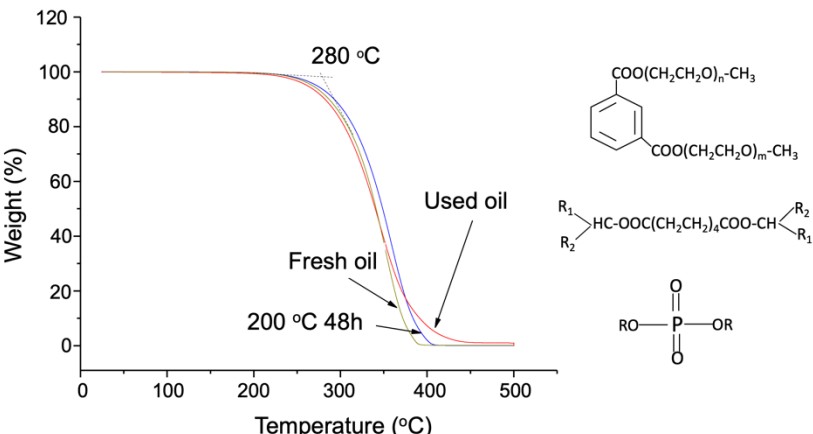

**Figure 2.** Thermogravimetric analysis of the three aerospace lubricant samples: the 200 °C, 48 h samples were prepared by heating the fresh oil capped in a glass vial, and the used oil was discharged from an aircraft engine system provided by Honeywell.

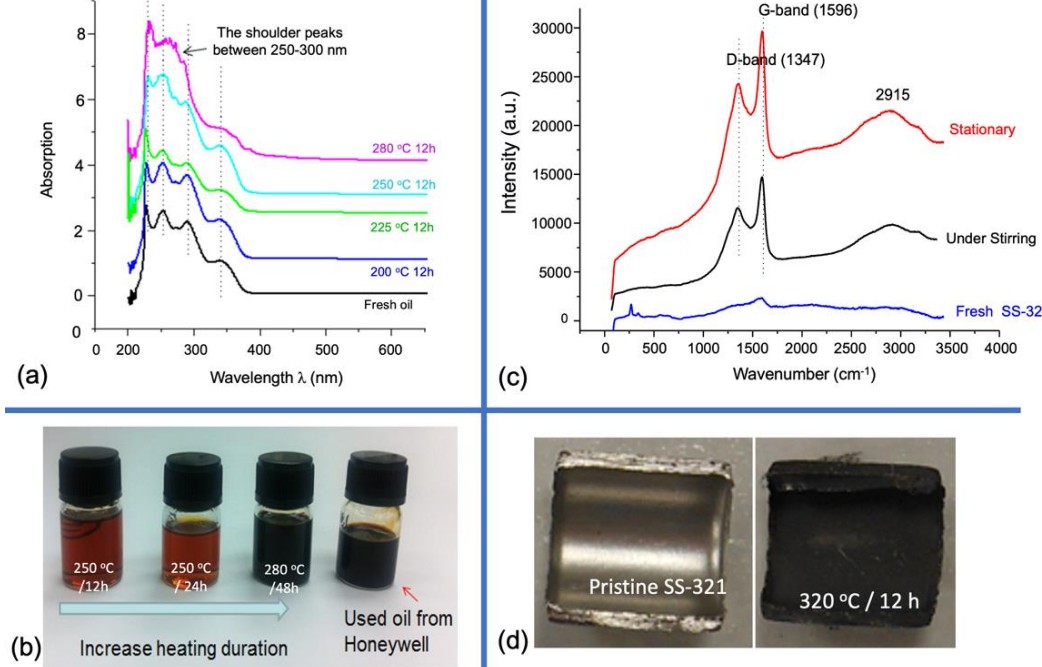

**Figure 3.** (**a**) The UV–Vis spectra of the aerospace lubricant samples (diluted in ethanol $10^{-3}$ $v/v$). The lubricants were treated in small glass jugs with different heating histories. (**b**) The photos of the oil samples show increasing thermal decomposition extent, in which the used lubricant oil was collected from the circulating pipe of an aircraft engine system as the reference. (**c**) Raman spectra of the inner surfaces after a new SS-321 tube were immersed in the lubricant at 320 °C for 12 h with and without stirring. (**d**) Photos of the pristine SS-321 and the stationary sample.

The UV–Vis spectra (Figure 3a) coincide with the TG analysis: heating at 280 °C for 12 h results in obvious derivatives exhibiting shoulder peaks between 250 and 300 nm, belonging to p→π* absorption of aromatics. In contrast, virgin oil has prominent absorption bands near the UV range, indicating aliphatic oxygen-containing groups as the main constituents suggested above. Moreover, the last two samples in Figure 3b present similar appearances, implying that heating at 280 °C for 48 h reaches the same severity as the actual heat soak in the aircraft engine system and impose on the lubricant. This test also

excludes the surface metal catalytic effect on coke formation as the heating was conducted in glass vials, reflecting only the intrinsic thermal vulnerability of the oil.

Except for the thermal decomposition in the oil bulk, coking over the interior surface of the SS-321 tube is dominant due to the surface entrapment, as illustrated in Figure 4 and the temperature gradient towards the tube axis. This perception was tested by immersing an SS-321 tube specimen in the lubricant oil in an autoclave reactor at 320 °C for 12 h. We observed that mechanical stirring in this experiment slightly helped impede coke development over the metal surface, according to Raman analysis (Figure 3c). The stationary sample has a more petite D to G peak height ratio (0.72) than the under-stirring sample (0.75), which signifies more extensive aromatic ring growth in the former. Mechanical stirring could constantly replace the surface oil layer and thus interfere with coking over the surface, although this effect is weaker compared to what temperature and surface defects bring about.

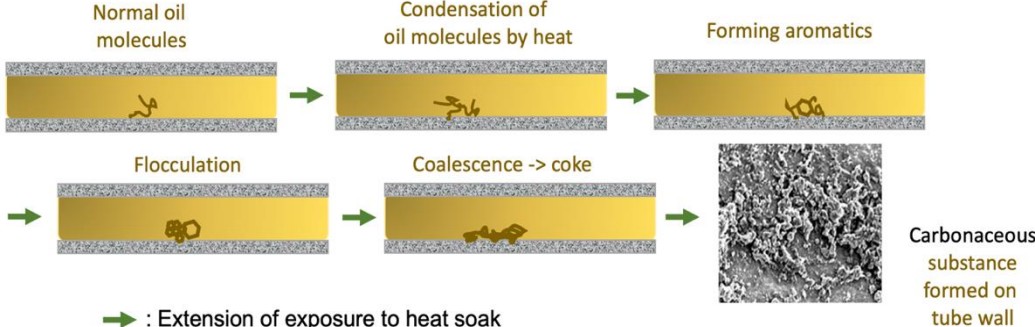

**Figure 4.** Schematic illustration of coke formation from thermal reactions of lubricant oil staying still in a metal tube in heat soak, where flocculation leads to fusion of organic aromatic flocs. The inset shows the deposition and growth of carbonaceous crumbs on the interior wall of SS-321 tubing.

Based on this observation, the presence of surface cracks over the crude inner SS-321 tube wall favors the development of coke (Figure 5a), as the microcracks and the coarse top oxide layer are more capable of entrapping organic aromatic floc. The fouling layer then expands by taking more suspended flocs. To validate this perception, as the first step, a new short SS-321 tube sealed with filled fresh lubricant was heated at 300 °C for 64 h in a furnace. As a result, a thick carbon layer was deposited over the inner tube wall (Figure 5b,c).

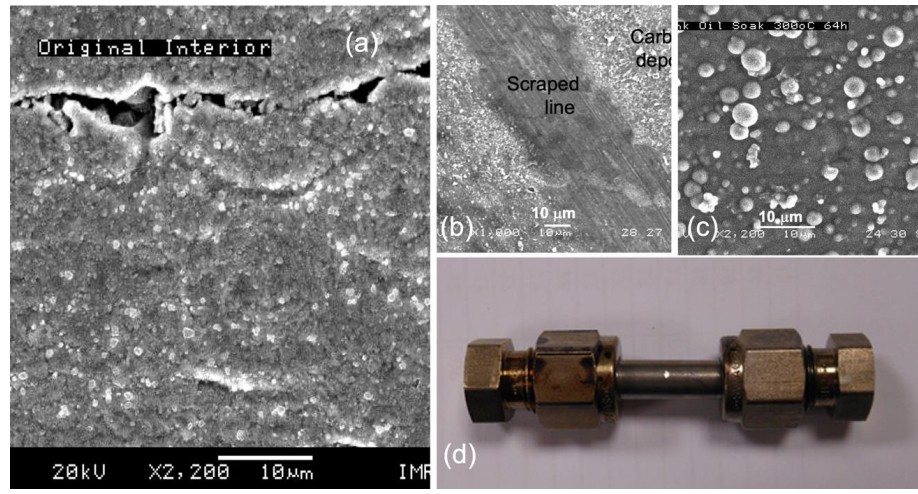

**Figure 5.** SEM images of (**a**) the interior tube wall of the crude SS-321 tube, (**b**) a coke layer deposited on the above surface after heating the lubricant sealed in the tube at 300 °C for 64 h, (**c**) an amplified view of the coke layer, and (**d**) the photo of the tube enclosing the aerospace lubricant.

### 3.2. Restructuring the Inner Tube Surface by a Chemical Etching Means Deferring Coking

On top of the conventional formulation of waterborne chemical etching solution [30], this work includes ion sequestering and stabilizing agents (Table 1).

**Table 1.** Variation in the surface roughness of the inner wall of the SS-321 tube with the change in etching conditions.

| Sample Tubes | Constituents of Etchant | | Etching Duration (min) | Average Surface Roughness (µm) |
|---|---|---|---|---|
| | $H_2O/H_2O_2/HCl/lactic$ Acid [c] vol. (mL) | E or N [d] (wt %) | | |
| C (crude) [a] | - | - | - | 6.44 |
| LHC [b] | 1/3/9/5 | 0 | 30 | 5.24 |
| LHC-E/a | As above | 0.57 | 10 | 6.52 |
| LHC-E/b | As above | 0.57 | 20 | 6.87 |
| LHC-E/c | As above | 0.57 | 30 | 6.28 |
| LHC-E | As above | 1.14 | 30 | 8.91 |
| LHC-N | As above | 0.14 | 30 | 5.40 |
| | $H_2O/H_2O_2/HCl/$ vol. (mL) | | | |
| SHP | 9/1.35/9 | | 30 | 1.46 |

[a.] The new SS-321 tube, i.e., tube C. [b.] To simplify the presentation, a sample 321 tube is named by the etchant used to treat the tube. [c.] The primary volume unit consists of the four solutions in various volumes. [d.] E: etidronic acid, N: 1,5-naphthalene disulfonic acid.

Many organic α-hydroxy acids and amino acids could fulfill the role of sequestering metal ions, which prevents hydrolysis of metal ions. We chose lactic acid as a ligand; along with its function to retard hydrolysis, it has far higher (coordination) stability constants with the nonferrous metal ions, e.g., $Cr^{3+}$, $Ni^{2+}$, $Mn^{2+}$, and $Cu^{2+}$, than with $Fe^{2+}$ [31,32]. The nonferrous metals constitute about 29–34 wt % of SS-321. Therefore, the chelating effect helps achieve the etching efficacy by enhancing the dissolution of more cathodic nonferrous metals such as Ni and Cu. Moreover, both stabilizing agents of E (etidronic acid) and N (1,5-naphthalene disulfonic acid) were identified to cast a specific topographic pattern (Figure 6) because both chemicals possess more robust surface (adsorption) activities and weaker hydrophilicity than lactic acid. Visually, LHC casts a spherulite contour (6b), whereas LHC-E creates a groove-like topography in each spherulite, and LHC-N lays a ridge-like outline.

The etching duration and dose of E shape the depth of streaks achieved. This claim refers to the LHC etchant that most significantly polishes the interior wall of the crude tube (tube C). Compared to the LHC-treated surface, adding 0.57% E raises the average roughness or mean depth of streaks, which varies with the extension of etching duration and settles at the least rough level (from/a to/c in Table 1) driven by the surface thermodynamic tendency. Nonetheless, the roughness is promoted by the increase in the dose of E from 0.57% to 1.14%, coinciding with the selective attaching of chelate E to the more reactive surface sites on the brinks once they are formed, as proposed above.

To examine how the inner tube roughness affects the thermal decomposition of the lubricant enclosed in the tube through a devised course of heating, the lubricant sample discharged after the test was analyzed by liquid chromatography (LC) to check the variation in composition. Moreover, the dynamic light analysis determines the carbonaceous particles (flocculent of aromatics) generated from heating. This inspection was limited to using LHC-E etchant with the variation in etching time and dosage of E (Table 2).

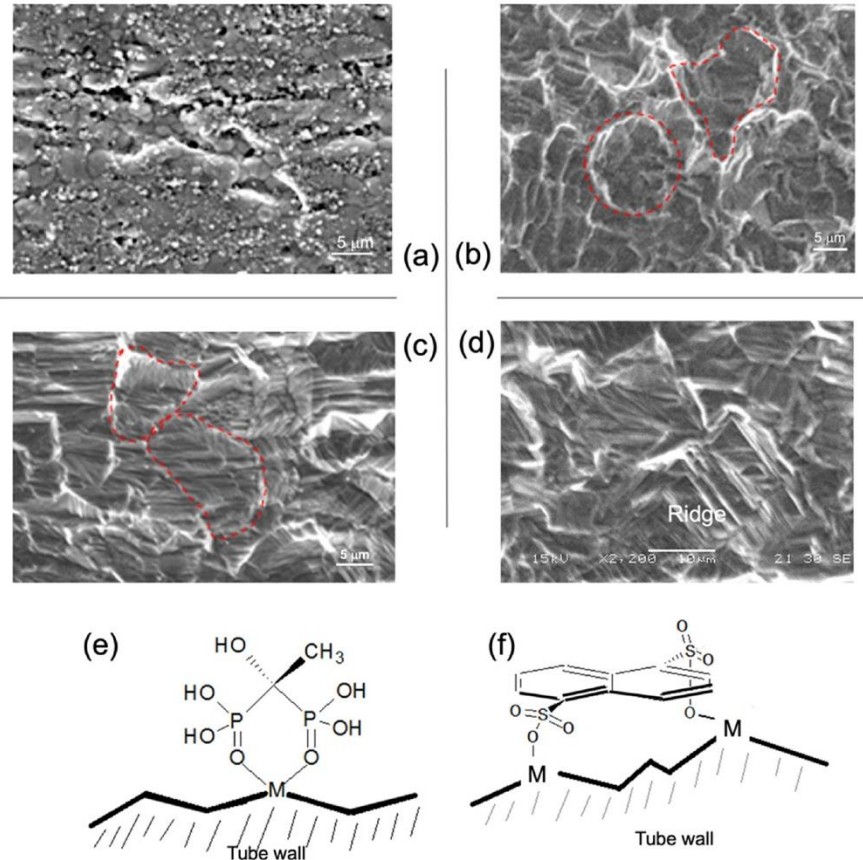

**Figure 6.** Electron microscopic images of the interior surfaces of SS321 tube: (**a**) crude; (**b**) by LHC; (**c**) by LHC-E; (**d**) by LHC-N. (**e**,**f**) The schematics show the stabilizing action of etidronic acid and 1,5-naphthalene disulfonic acid, respectively. The scale bars are 5 μm in (**a**) to (**c**), and 10 μm in (**d**).

**Table 2.** The relative composition of a lubricant sample, based on LC analysis, discharged from an SS-321 tube treated by an etchant after the devised heating course.

| Oil-Samples [a] | | Retention Time (min) of LC Peaks [c] | | | | | | | | |
|---|---|---|---|---|---|---|---|---|---|---|
| | | 1.81 | 2.22 | 3.49 | 3.71 | 3.84 | 4.13 | 4.55 | 5.68 | 8.55 |
| Fresh | | 100 [b.] | 4.54 | | 12.1 | | | | | |
| C (crude) | | 100 | 10 | 7.1 | 9.2 | 10 | 7.9 | 2.1 | 1.1 | 0.42 |
| LHC-E/a | Relative peak | 100 | 8.1 | 6.7 | 9.4 | 7.6 | 7.2 | 2.3 | 0.9 | 0.45 |
| LHC-E/b | intensity (%) [b] | 100 | 8.4 | 6.7 | 8.9 | 8.4 | 7.6 | 2.2 | 0.88 | 0.44 |
| LHC-E/c | | 100 | 8.5 | 6.2 | 8.5 | 7.9 | 7.4 | 1.7 | 1.1 | 0.56 |
| LHC-E | | 100 | 8.9 | 5 | 8.9 | 8.9 | 5.6 | 1.7 | 1.1 | 0.56 |

[a] Samples are the lubricants discharged from the treated SS-321 sample tubes defined in Table 1. The lubricants were prepared (Section 2.3) in two continuous cycles without changing the oil. [b] For each sample, the peak intensity of the highest peak of LC is set to be the denominator to show the respective contents of the other main constituents. [c] The Supplementary Materials (Figure S2) present the selected LC diagrams.

The LC peaks of all the tested oil samples are tableted against those of the new lubricant. The comparison unveils the following points. (i) Heat soaking at 300 °C for 60 h brings about six leading derivatives, primarily due to the decomposition of the primary component of the fresh lubricant (having retention time at 1.81 min) and the second component appearing at 3.71 min. (ii) The primary component in tube C undergoes the most significant extent of decomposition that generates the most significant contents of the high-molecular-weight (HMW) species with LC column retention times at 3.84 min

and 4.13 min, respectively. These two components are much more concentrated than the other three species having longer retention times. (iii) Among the three sample tubes etched by LHC-E/(a to c), sample LHC-E/b, being the roughest, gives rise to the highest contents of the two HMW derivatives, as indicated above. (iv) A significant increase in average roughness in LHC-E relative to LHC-E/(a to c) results in a higher content of HMW derivatives at 3.84 min, but a smaller content of the HMW species at 4.13 min.

On the other hand, the mean carbonaceous particle size and standard deviation of each tested oil sample were analyzed (Table 3). The outcomes are in parallel with the above LC analysis results. (i) The oil drained from sample tube C contained the largest and most broadly distributed particles, which were formed from the soluble HMW species detected by LC. (ii) Tube LHC-E acquired larger particles than LHC-E/(a to c) and a slightly wider particle size distribution. From the perspective of the reaction mechanism, the rougher the topographic feature over the dense surface, the more favorable the adsorption of the HMW derivatives will be. Hence, the more aromatic floc is to be formed on the inner tube wall because of the adsorption capability of rough spots and the temperature gradient from the tube wall to its axis.

**Table 3.** The particle sizes of aromatic flocculent suspended in the lubricant after the heating test [a].

| Etchant No. | Mean Particle Size (μm) | Standard Deviation (μm) |
|:---:|:---:|:---:|
| C | 10.9 | 10.5 |
| LHC-E/a | 0.49 | 0.04 |
| LHC-E/b | 0.63 | 0.18 |
| LHC-E/c | 0.47 | 0.03 |
| LHC-E | 1.13 | 0.21 |

[a]. Samples as listed in Table 2. Refer to Section 2.4 for the technique to carry out the analysis.

Figure 7 shows the corresponding outcomes of the above inference. Firstly, the inner surface of tube C was entirely covered by a coke layer, but tube LHC-E experienced minor coking (Figure 7a,c) after a five-cycle (12 h/cycle) heating treatment. Secondly, the inner tube surface of both LHC-E and LHC-E/c after a heating cycle (56 h) with a prolonged dwelling time (without changing oil) presented slightly different coking extents, and the former incites slightly larger coke particles than the latter (Figure 7d,e). This could be attributed to the effect of roughness that encourages the growth of coke grains.

Further differentiating the difference between tube C and tube LHC-E as well as the difference between tube LHC-E and tube LHC-E/c in terms of their different coking extents, we found that the dosage and the stabilizing agent influence surface roughness (Figure 6b–d). LHC-N attains a lower surface roughness than LHC-E/c and LHC-E (Table 1) because of a far smaller dose of N added. The sulfonic acid group is more potent than the phosphoric acid group in binding metals. This quantity of N reflects the optimal level to achieve the most vital coke prohibition (presented below). It has been proposed above that other than roughness, the topographic pattern affects coke development on the tube wall. LHC-N exhibits a ridge-like topography different from the grooved spherulite topography of LHC-E.

We conducted two tests to investigate the coke deposition situation on the inner tube wall of these two samples: (1) the single heating cycle (56 h) and (2) the three consecutive heating cycles (3 × 56 h) in which the oil was changed before starting a new cycle. As for the first test, we observed that similar grain sizes and number densities happen in both LHC-N and LHC, which have similar surface roughness values, whereas LHC-E presents a smaller concentration of the distributed coke particle than the above two according to their SEM images of relatively lower amplification (Figure 8). However, more detailed scrutiny of these three coked surface reveals that contrary to LHC and LHC-N, LHC-E contains many tiny coke grains of a few dozen nanometers that are grown on the grooved patch of each spherulite (Figure S3).

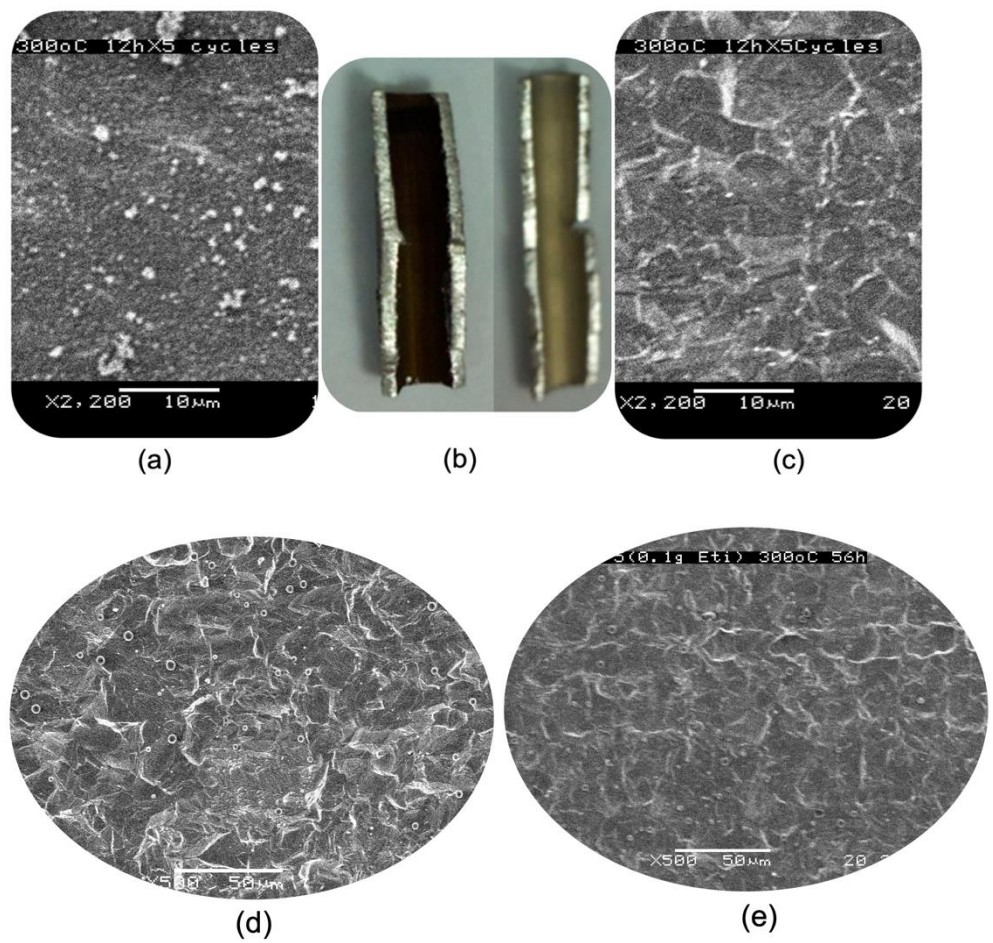

**Figure 7.** Examination of coking extents occurring in two sets of selected sample tubes: (**a**) the SEM image of tube C and (**c**) that of tube LHC-E after the five-heating cycle (5 × 12 h) test (Section 2.3) and (**b**) the photos of the above two samples; (**d**) the SEM image of sample tube LHC-E and (**e**) that of LHC-E/c after a single heating cycle at 300 °C for 56 h. The scale bars are 50 μm in (**d**,**e**).

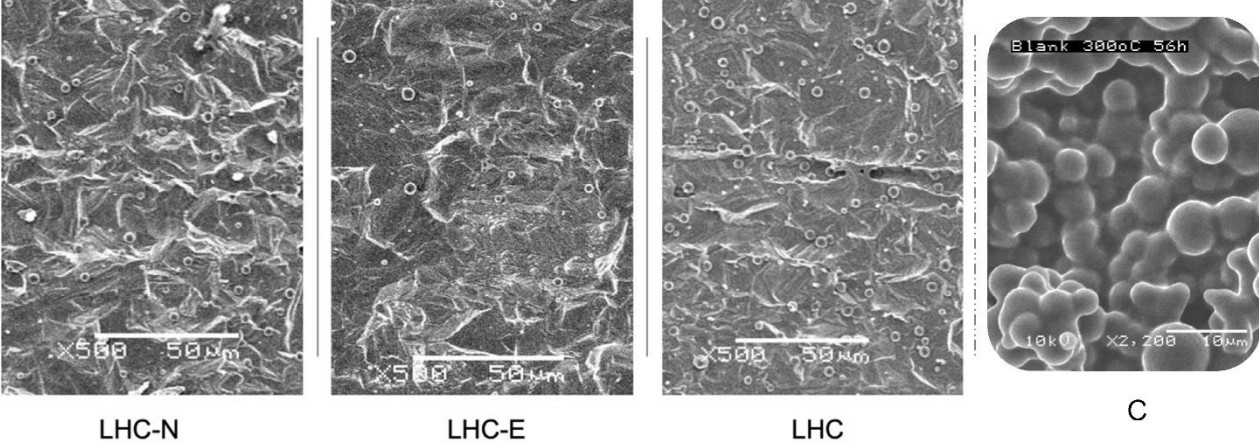

**Figure 8.** Coking extents inside the three sample tubes vs. the crude tube after a cycle (56 h) of heating sealed lubricant at 300 °C. The scale bars are 50 μm in the left three images and 10 μm in C.

As for the quantitative comparison, the coke developed in the above three samples (Figure 8) is too light to be determined by TG analysis as the tubing substrate takes up the dominant mass fraction. However, tube C, the crude sample, exhibits much heavier coking than all the above three samples according to its surface morphology. Regarding

the second test, increasing the testing severity results in somewhat different outcomes, as summarized in Table 4. A careful sample preparation for the quantification was carried out: the four sample tubes were immersed in a mixture of acetone–ethanol ($v/v = 1$) for a few rounds of ultrasonicating to remove soluble species after the coking test, and then dried. The cokes deposited were, therefore, obtained by the gravimetric analysis. The highest coking deposition was in tube LHC-E, except for tube C.

**Table 4.** Comparison of the coke deposition through a 3-cycle (56 h/cycle) oil heating test.

| C | LHC | LHC-N | LHC-E |
|---|---|---|---|
| 0.708% | 0.044% | 0.028% | 0.288% |

Regarding the regular chemical steel etchant, SHP represents a standard formulation of aquatic etchant and was selected as the control to compare the LHC etchant. Using SHP leaves a very different topography from what is achieved by LHC (Figure S3 and Figure 4) and a rougher inner tube wall. Consequently, the inner wall of the SHP tube is overlaid with coke (Figure S4). The outcome indicates that the SHP-etched surface cannot improve the coke resistance of tube C. The main difference between these two designs of etchant is the presence of lactic acid in LHC, in which the higher concentration of HCl does not result in a rougher surface. In principle, lactic acid behaves somewhat like a corrosion inhibitor, retarding the anodic reaction of ferrous metals by HCl. The detailed interfacial corrosion and inhibition mechanisms are discussed in a recent review article [33]. The selective non-ferrous metal ion sequestering role of lactic acid assists the dissolving of the nonferrous metals and thus lightens iron removal from tube C. The most critical function of lactic acid lies in its proficiency in crafting a groove-like pattern to offer the aforementioned fluid agitation effect.

*3.3. The Role of the Axial Shear Flow of the LHC-Based Etchant in the Attainment of the Surface Microstructures*

In addition to the recipe and etching duration (Table 1), the flow rate of etchant liquid and dosage per length of sample tube affect the topography and roughness of the inner tube wall. The coking extent on the inner tube wall is therefore apparently impacted by varying the latter two factors. Regarding the flow rate, the previous samples were obtained by setting the peristaltic pumping rate at 150 rpm (Figure 1 and Section 2.2). When the liquid delivery rate was reduced to 50 rpm, coking becomes heavier in the tubes etched by LHC-N and LHC compared to their counterparts prepared by the designated flow rate (Figure 9). In contrast, LHC-E could maintain the same coking extent as when 150 rpm is applied (Figure 5). This outcome is attributed to the insufficient engraving extents happening in sample tubes LHC and LHC-N so that the topographies left behind cannot stimulate essential surface turbulence fields. Nevertheless, LHC-E can still attain the required surface pattern for impeding the development of coke particles because of the impact of component E. To further verify the leverage of the axial flow rate of the etchant on coking, we designed an immersing protocol to test the three different etchants. As expected, the static etching led to heavy coke coverage in the three sample tubes (Figure S5). Alternatively, raising the pumping rate was also probed; we used LHC-E to conduct the test, for which a 250 rpm pumping rate was tested. Although this flow rate increases the surface roughness, the adsorption overtakes the surface microflow agitation, leaving behind a far heavier coking layer than observed in the tube prepared by the immersion etching (Figure S6).

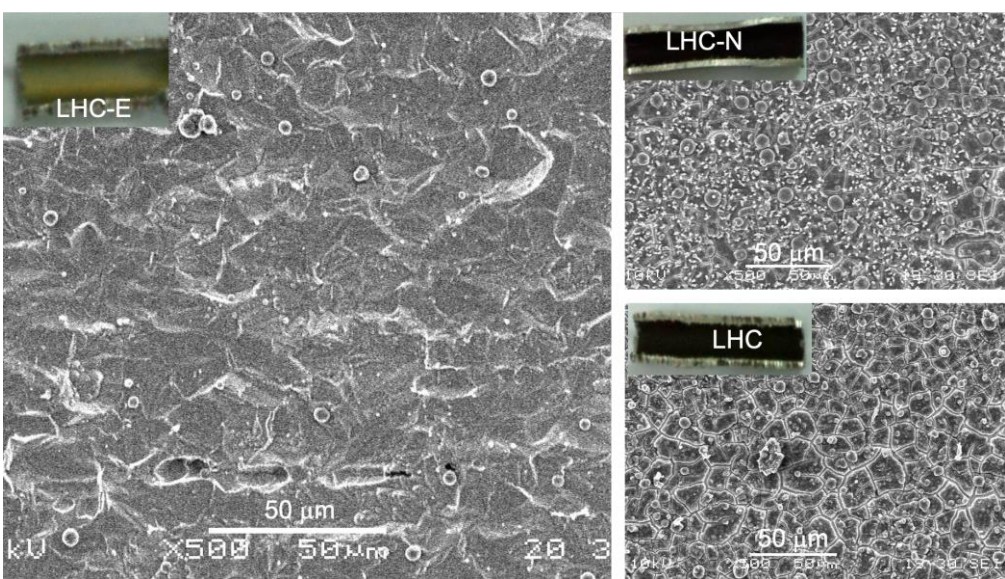

**Figure 9.** Examination of coking extents occurring inside the three sample tubes that were prepared using a 50 rpm pumping rate to circulate the etchants after a single cycle. Before performing a microscopic check, the sealed lubricant was subjected to a single heating cycle (56 h) at 300 °C. (Insets: photos of the scissored tubes.)

Following the above context, the impact of etching dosage on the coke-inhibiting property was also tested. The other conditions remained unchanged, including the 30 min etching duration, 150 rpm pumping rate, and 9 cm tube. Referring to the typical dosage of LHC-E that successfully mitigates the coking in tube C (Figure 6), applying one-third and two-thirds of it shows the consequence of inadequate etching (Figure S6). The sample prepared using two-thirds of the total dose of LHC-E could still attain a limited coking extent. Finally, it is essential to understand how the etching of the crude SS-321 tube alters its mechanical properties. The LHC-E sample tube was selected for the assessment (Table 5). The etching causes about a 7% decrease in tube thickness and a slight loss of the modulus of elasticity but a small increase in tensile strength. It reflects the effect of polishing away pits and microcracks distributed on the inner wall of the crude tube wall since these loci are vulnerable to causing defects during the tension test.

**Table 5.** Testing of tension before and after chemical etching.

| Property | Crude Tube | LHC-E Sample Tube |
|---|---|---|
| Wall thickness (mm) | 0.88 | 0.82 |
| Tensile strength (MPa) | 679.4 | 687.4 |
| Modulus of elasticity (GPa) | 104.7 | 102.5 |

## 4. Discussion

### 4.1. Coking Mechanism at the Interface between Oil and Steel

Following the illustration in Figure 4, the coking is self-sustained once a spread of carbonaceous grains happens over the tube wall; these initial grains then proliferate the deposition of carbonaceous substance by immobilizing more HMW derivatives formed in the bulk of oil, as well as surrounding them while the oil is being heated continuously. Given this, to inhibit coking in the SS-321 tube, making the inner wall incapable of lodging aromatic flocs is essential, which logically should involve removing or covering up the existing pits, microcracks, and the top coarse oxide layer inside the crude tube. Therefore, equipping the inner tube surface with a structure-based rejection mechanism to HMW derivatives is crucial. As the flocs are soft coils of hydrocarbon, the fluid eddy flow layer

overlying the inner tube wall is proper to suspend the flocs. Further aromatization via condensation of the flocs can thus be significantly lessened.

### 4.2. The Surface Chelating Effect on Crafting Microstructures and the Subsequent Leverage on Coking

In Figure 6, the two fine surface topographies (c–d) contain brinks that originate from the selective chelation of either E or N with the metal atoms located at the sharp positions (as illustrated in e–f) since these metal atoms have lower coordination numbers and are more reactive to captivate E or N. As a result, those having higher coordination numbers, e.g., located on terraces described by the terrace ledge kink (TLK) model [34], could be more readily etched so that the delicate 3D topographic patterns were left behind. Indeed, these three topographies (b–d) reveal somewhat different capabilities to defer the growth of a carbonaceous layer. We demonstrate this point later. The EDS spectroscopy verified the proposal regarding the role of a stabilizing agent (Table S1). It shows that the contents of nonferrous metal elements decrease with the increase in the dose of stabilizing agent E. This can be interpreted as the selective capping of E to sharp surface Fe sites, namely highly coordinatively unsaturated sites. A further increase in E to 2.3 wt % brings about a significant increase in C%, indicating that more surface Fe sites and even Ni sites are covered up by E. Correspondingly, the variation in the surface roughness of the inner tube wall by LHC and the LHC-E series (Table 1) is in accord with what Figure 6a–c exhibits.

The topographic pattern affects thermal decomposition routes except for surface roughness. Typically, although tube C has a lower roughness level than tube LHC-E, it incites higher contents of derivatives of aromatic flocs. There are, however, no pits, crevices, or a coarse oxide layer in tube LHC-E to entrap HMW derivatives and their flocculation species. On the contrary, the surface entrapment occurs in tube C, as mentioned above.

Regarding Table 4, the outcome originates from a spread of nano coke particles over each grooved spherulite domain. In contrast, tube LHC, having a shallower topographic pattern, has an ineffective adsorption capability to hold organic flocs and to allow a proliferation of them on the inner tube wall. On top of the surface structural influence, the change in lubricating oil after finishing the first two cycles also favors hose organic flocs attached to the surface to move into the oil phase, affecting the final amount of coke staying in the samples. Another important observation is the ridge-like topography of LHC-N, which manifests apparent resistance against coke development over its surface compared to, in particular, tube LHC-E. Fundamentally, two pivotal surface effects are affecting the coke-forbidding capability of these three samples: (1) the number of sharp sites, since they encourage adsorption and coke development, and these sites could more retain organic flocs once the oil is changed, and (2) the efficacy of the topographic pattern to induce a layer of microturbulence fields overlying the surface, which is caused by the frequent alternations in heat flux in at a micron-scale overlying the inner tube wall, as described in the Introduction. Both effects drive opposite changes in coking because the second effect imposes a micro-agitation result to push back organic flocs to the bulk of the oil. Thus, the ridge-like topography of tube LHC-N offers a more substantial eddy agitation effect than the grooved spherulite topography in tube LHC-E. As for the surface of tube LHC, it possesses weaker surface entrapment but could still maintain an eddy liquid layer over it.

### 5. Conclusions

We developed a solution to mitigate coking in a stainless steel (SS)-321 tube (id = 4.7 mm). The solution counts on an enhanced chemical etching system, including recipe development and application. The study leads to the following five conclusions:

i.     The coke formation in the SS-321 tube enclosing aerospace lubricant involves the construction of high-molecular-weight organic flocs and their entrapment on the inner surface of the crude tube, preferentially at tiny cracks, where the flocs are readily converted to carbonaceous grains, which by themselves also proliferate coke growth.

ii.    Formulation of an alternative type of etching solution consisting of the basic formula, lactic acid-HCl-$H_2O_2$-$H_2O$ (LHC), plus etidronic acid (E) or 1,5-naphthalene disulfonic acid (N) as a stabilizing agent.

iii.    Carrying out chemical etching using the three etchants, LHC, LHC-E, and LHC-N, through the axial flow to treat the SS-321 tube, which removes original defects and casts two specific topographic patterns at a micron-scale on the inner tube wall. In particular, LHC and LHC-E result in a grooved spherulite topographic pattern, whereas LHC-N has a ridge-like topographic pattern.

iv.    Examination of the leverages of surface roughness and topography on coking shows that removing micro defects and the oxide layer from the inner surface of the crude tube significantly enhances coke resistance. The roughness of the topography encourages the surface entrapment, but either topographic pattern counters the surface entrapment. The latter effect is attributed to micro-turbulence fields. Moreover, the ridge-like topography demonstrates a more extended anti-coking performance than the other pattern.

v.    The critical etching conditions affecting the anti-coking attributes include dosage per tube length, the optimal pumping rate, and etching duration concerning an etching recipe.

The coking resistance properties, sustained by the two chemically craved steel surface patterns, originate from rejecting the deposition of hydrocarbon flocs on the surface.

**Supplementary Materials:** The following supporting information can be downloaded at: https://www.mdpi.com/article/10.3390/jcs6090266/s1, Table S1: EDS data of the polished internal wall of SS-321 tube with the variation in etidronic acid (E) dose in LHC-E chemical polishing formula. Figure S1: The FT-IR spectrum of the fresh lubricant (Section 2.1), in which the absorption bands are attributed to the respective bond vibration types. Figure S2: The selected liquid chromatography diagrams of the two samples, where the insets provide the detailed peaks of the main components. The other samples have a similar diagram as sample E4. Figure S3: SEM images showing the detailed inner wall surface morphologies of the three tested tubes in question that were tested by heating the sealed lubricant at 300 °C for a course of 56 h. The arrow signs in the left image indicate numerous cokes over spherulites. Figure S4: SEM image of the SS-321 tube etched by SHP and that of the tested tube after going through a single oil-heating cycle (56 h) test at 300 °C. Figure S5: Examination of coking extents occurring inside the three sample tubes that were prepared by immersing in the same dose of etchant for 240 min. The treated tubes were subjected to a single heating cycle (56 h) at 300 °C before performing the microscopic check. Figure S6: Leverages of the different etching conditions on the coking extent in the treated tubes.

**Author Contributions:** Conceptualization, Y.Z., S.W.T. and L.H.; methodology, Y.Z.; formal analysis, S.W.T.; investigation, Y.Z., S.W.T. and L.H.; resources, Y.Z. and L.H.; data curation, Y.Z.; writing—original draft preparation, Y.Z. and S.W.T.; writing—review and editing, L.H.; visualization, S.W.T.; supervision, L.H.; project administration, S.W.T. and L.H.; funding acquisition, L.H. All authors have read and agreed to the published version of the manuscript.

**Funding:** This research was funded by A*Star Aerospace Programme for providing financial support (SERC grant no. 112 155 0602) (https://www.a-star.edu.sg/aerospace (accessed on 25 January 2013)).

**Data Availability Statement:** Data in this article and supplementary material were solely obtained from our laboratory experiments.

**Acknowledgments:** We thank the A*Star Aerospace Programme for funding this research project. We are also grateful to Honeywell Inc. for providing industrial perspectives, new steel tubes, and aerospace and lubricating for this project.

**Conflicts of Interest:** The authors declare no conflict of interest. The funders had no role in the design of the study, in the collection, analyses, or interpretation of data, in the writing of the manuscript, or in the decision to publish the results.

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
