# Peer review of "Crafting Metal Surface Morphology to Prevent Formation of the Carbon–Steel Interfacial Composite"

_jcs, doi:10.3390/jcs6090266_

Round 1
Reviewer 1 Report
The authors submitted "Crafting Metal Surface Morphology to Prevent Formation of the Carbon-Steel Interfacial Composite" for publication in "J. Compos. Sci" the novelty of this work is good.
1. The schematic scheme of the material's chemical structure should be provided.
2. FTIR analyses should be redone. these data are unacceptable.
3. The scale bar in some images is unclear.
4. The conclusion part should be summarized.
5. How about the mechanical properties?
6. some references should be provided such as Macromolecular Chemistry and Physics 220, 2019, 1800306, doi.org/10.3390/ma15062023.
Author Response
Reviewer-1
Comment-1: The schematic scheme of the material's chemical structure should be provided.
Response:
Thanks to this recommendation, the four key components’ structures are listed in Figure 1.
Comment-2: FTIR analyses should be redone. these data are unacceptable.
Response:
Thanks for bringing up this concern. The first version of the manuscript mentioned the three main types of compounds consisting of the lubricant in the first paragraph of Section 3-1, but it did not include an FTIR spectrum of the lubricant to correlate with the description of the oil composition.
To follow the reviewer’s comment, we add the FTIR spectrum as a supplementary fact (Fig. S-1) in this revision. In addition, we have elaborated on this new addition in lines 193-195.
Comment-3: The scale bar in some images is unclear.
Response:
Thanks for pointing out this weak aspect. We have modified the scale bar labels in Figs. 5 and Fig. 9, respectively, and specified the lengths of scale bars in the captions for Figs. 6 to 8.
Comment-4: The conclusion part should be summarized.
Response:
Thanks for this good point. A brief remark has been added at the end of the conclusion.
Comment-5: How about the mechanical properties?
Response:
The second part in the last paragraph under the Results section has reported the leverage of chemical etching on the mechanical properties of the sample tubes. The changes in Young’s modulus and Tensile strength of the representative treated sample are compared with the crude tube in Table 5.
Comment-6: some references should be provided, such as Macromolecular Chemistry and Physics 220, 2019, 1800306, doi.org/10.3390/ma15062023.
Response:
Thanks for this suggestion. This reference is “Functional Silica and Carbon Nanocomposites Based on Polybenzoxazines” in Volume 220, Issue 1 1800306 Macromolecular Chemistry and Physics. However, the paper has a different DOI, https://doi.org/10.1002/macp.201800306, from what is given in the above comment. This article has, unfortunately, little to do with our present work.
The above DOI is, however, a review article about the corrosion inhibitors for carbon steel in 1.0 M HCl. We cite this paper in the revised version (ref. 33) because it is related to our work in terms of the role of lactic acid.
Reviewer 2 Report
The paper is interesting and the results can be important for aircrafts engines users. I suggest that in the introduction the problem of coke formation in steel pipes should be described in more detail - the scale of the problem and the importance for the operation of aircraft engines.
Author Response
Reviewer-2
The paper is interesting and the results can be important for aircrafts engines users. I suggest that in the introduction the problem of coke formation in steel pipes should be described in more detail - the scale of the problem and the importance for the operation of aircraft engines.
Response:
Thanks for this comment. We have embedded a short description in the 2nd paragraph (lines 58-65) of the Introduction to cover the concerns of today’s aviation industry about the coking issue in the airplane engine oil system.
Round 2
Reviewer 1 Report
This manuscript could be published in this journal.